


# Impact of bioenergy crops expansion on climate-carbon cycle feedbacks in overshoot scenarios

Irina Melnikova[1,2], Olivier Boucher[1], Patricia Cadule[1], Katsumasa Tanaka[2,3], Thomas Gasser[4], Tomohiro Hajima[5], Yann Quilcaille[6], Hideo Shiogama[3], Roland Séférian[7], Kaoru Tachiiri[3,5], Nicolas Vuichard[2], Tokuta Yokohata[3] and Philippe Ciais[2]

[1]Institut Pierre-Simon Laplace (IPSL), Sorbonne Université / CNRS, Paris, France
[2]Laboratoire des Sciences du Climat et de l'Environnement (LSCE), IPSL, Commissariat à l'énergie atomique et aux énergies alternatives (CEA/ CNRS/ UVSQ), Université Paris-Saclay, Gif-sur-Yvette, France
[3]Earth System Division, National Institute for Environmental Studies (NIES), Tsukuba, Japan
[4]International Institute for Applied Systems Analysis (IIASA), Vienna, Austria
[5]Research Institute for Global Change, Japan Agency for Marine-Earth Science and Technology, Kanazawa-ku, Japan
[6]Institute for Atmospheric and Climate Science, ETH Zürich, Switzerland
[7]CNRM, Université de Toulouse, Météo-France, CNRS, Toulouse, France

*Correspondence to*: Irina Melnikova (irina.melnikova@lsce.ipsl.fr)

**Abstract.** Stringent mitigation pathways frame the deployment of second-generation bioenergy crops combined with Carbon Capture and Storage (CCS) to generate negative $CO_2$ emissions. This Bioenergy with CCS (BECCS) technology facilitates the achievement of the long-term temperature goal of the Paris Agreement. Here, we use five state-of-the-art Earth System models (ESMs) to explore the consequences of large-scale BECCS deployment on the carbon cycle and carbon-climate feedback under the CMIP6 SSP5-3.4-OS overshoot scenario, keeping in mind that all these models use generic crop vegetation to simulate BECCS crops. We show that an extensive cropland expansion for BECCS causes ecosystem carbon loss that drives the acceleration of carbon turnover and affects the estimates of the absolute values of the global carbon-concentration $\beta$ and carbon-climate $\gamma$ feedback parameters. Both parameters decrease so that global $\beta$ becomes less positive, and $\gamma$ – more negative. Over the 2000–2100 period, the land-use change (LUC) for BECCS leads to an offset of the $\beta$-driven carbon uptake by 12.2% and amplifies the $\gamma$-driven carbon loss by 14.6%. A human choice on land area allocation for energy crops should take into account not only the potential amount of the bioenergy yield but also the LUC emissions, and the associated loss of future potential change in the carbon uptake via the $\beta$ and $\gamma$ feedbacks. The dependency of the estimates of $\beta$ and $\gamma$ on LUC is very strong after the middle of the 21st century in the SSP5-3.4-OS scenario but it also affects other SSP scenarios and should be taken into account by the integrated assessment modelling teams and accounted for in mitigation policies so as to limit the reductions of the $CO_2$ fertilization effect where BECCS or land use expansion of short vegetation is applied.



## 1 Introduction

All stringent future socio-economic mitigation scenarios have negative emissions that rely on carbon dioxide removal (CDR) technologies (Fuss et al., 2014, Rogelj et. al., 218). CDR is important especially in overshoot scenarios, in which temperature temporarily exceeds the given target, e.g., the Paris Agreement temperature target, before ramping down as $CO_2$ is withdrawn artificially from the atmosphere (Jones et al., 2016a; Keller et al., 2018; Tanaka et al., 2021).

Bioenergy with Carbon Capture and Storage (BECCS) is one of the most cost-effective CDR technologies (Jones and Albanito, 2020; Babin et al., 2021). In BECCS, atmospheric $CO_2$ is captured via photosynthesis and fixed into plant biomass. Harvested biomass is then converted into bioenergy or directly combusted and a fraction of the carbon contained in the $CO_2$ produced is recuperated and is stored in geological reservoirs without being released back to the atmosphere (Canadell and Schulze, 2014). BECCS is a nascent CDR technology that has not been proven at large spatial scales. Its potential advantages include technical feasibility and a relatively low discounted cost in future decades that allows spreading mitigation efforts over a longer period (Anderson and Peters, 2016; Dooley et al., 2018).

The limitations of BECCS are the requirement of potentially large land areas, a loss of biodiversity, and the need for extra water and nutrients (Heck et al., 2018; Séférian et al., 2018; Li et al., 2021). Besides, BECCS may lead to a large amount of carbon emissions from land-use change (LUC), when bioenergy crops are grown over high-carbon content ecosystems such as grassland and forest (Clair et al., 2008; Gibbs et al., 2008; Schueler et al., 2013; Smith et al., 2016; Harper et al., 2018; Whitaker et al., 2018). The LUC emissions released due to land conversion to bioenergy crops include immediate (direct) greenhouse gas (GHG) emissions associated with the destruction of biomass and slash during LUC but also delayed (indirect) emissions from the decay of stumps and soil carbon. These emissions are termed as "carbon debt" (Clair et al., 2008; Fargione et al., 2008; Gibbs et al., 2008; Krause et al., 2018) because for BECCS to be carbon neutral, this loss of carbon must be paid back by several cycles of BECCS harvest followed by carbon geological storage, assumed to substitute with fossil carbon emissions. Using low-productivity marginal or degraded lands for the deployment of second-generation bioenergy crops (such as miscanthus or switchgrass) reduces the carbon debt because such lands have less carbon to lose. Further, soil carbon sequestration, in the long run, may even be achieved with BECCS if non-harvested residues of BECCS crops exceed the carbon input to the soil of the native ecosystems they substitute (Campbell et al., 2008; Gibbs et al., 2008; Mohr and Raman, 2013; Whitaker et al., 2018).

The issue with putting second-generation bioenergy crops in low-productivity lands is a need to invest large areas of land (Jones et al., 2016a; Smith et al., 2016). Currently, some land ecosystems act as a carbon sink primarily driven by the $CO_2$ fertilization effect on photosynthesis and the carbon turnover in ecosystems, which is often expressed as the carbon-concentration ($\beta$) feedback, although it is partly counterbalanced by the carbon-climate ($\gamma$) feedback (Jones et al., 2016b; Friedlingstein et al., 2020) which expresses the loss of ecosystem carbon per unit of global warming. The $\beta$ and $\gamma$ feedback parameters refer to the changes in the ecosystem carbon storage relative to the changes in the atmospheric $CO_2$ concentration ($\Delta CO_2$) and global surface air temperature (GSAT, $\Delta T$), respectively, relative to the pre-industrial level so that the changes in the carbon storage can be decomposed into the $\beta$ and $\gamma$ contributions ($\beta \times \Delta CO2$ and $\gamma \times \Delta T$, respectively). Here the temperature change is taken as a proxy for the response of the ecosystem carbon storage to climate change. As croplands, unlike other ecosystems, have limited potential to store additional carbon because the biomass is harvested regularly, and as the new





croplands have a lower soil carbon stock with a short turnover time for soil carbon, the large-scale BECCS
deployment must affect the land carbon cycle β and γ feedback parameters, although this has not been specifically
looked at in Earth System Models (ESMs) simulation results. Conventionally, β and γ are estimated at a global
scale and are assumed to be responses of "natural" ecosystems to the changes in $CO_2$ and climate, so that the
effects of LUC on these parameters are overlooked. No study to date has estimated the effects of BECCS
deployment on the terrestrial carbon cycle feedback parameters under overshoot and other scenarios.
In this study, we estimate the impact of large-scale BECCS deployment on the carbon cycle feedbacks under the
Shared Socioeconomic Pathway (SSP) overshoot scenario named SSP5-3.4-OS that includes mitigation policies
via an increase in the land area covered by second-generation bioenergy crops for CDR (Hurtt et al., 2020). We
use simulations from five Coupled Model Intercomparison Project 6 (CMIP6) ESMs to decompose the global β
and γ contributions to the changes in land carbon pools in ecosystems with and without LUC effects.
**2 Data and methods**
**2.1 SSP5-3.4-OS scenario**
The SSP5-3.4-OS follows the high-emission SSP5-8.5 scenario and branches from it in 2040 when aggressive
mitigation policies are implemented (O'Neill et al., 2016; Meinshausen et al., 2020). The delayed mitigation leads
to an overshoot of the Paris Agreement 2 °C temperature limit. In addition to a decline in fossil fuel emissions,
mitigation efforts after 2040 include the expansion of second-generation bioenergy crops (for BECCS) at the cost
mainly of pasture lands (Hurtt et al., 2020). There is no deforestation assumed after 2010, in order to preserve the
areas with high carbon content. Second-generation bioenergy crops account for most of the new cropland areas
deployed after 2040. In addition, a part of the existing croplands is converted to BECCS (Figure S1).
**2.2 CMIP6 ESMs**
We use five CMIP6 ESMs that simulate the SSP5-3.4-OS (Table 1). In addition to fully coupled simulations
(COU), biogeochemically (BGC) coupled simulations, where only changes in the atmospheric $CO_2$ concentration,
and not the temperature, affect the carbon-cycle processes, are also provided as part of the Coupled Climate–
Carbon Cycle Model Intercomparison Project (C4MIP) (Jones et al., 2016b). The combination of COU and BGC
simulations allows us to study carbon cycle feedback parameters.
The LUC emissions in the ESMs can be estimated as the difference in net biome production (NBP) between
simulations with and without land-use change that is between the "historical" and "hist-noLu" simulations for the
historical period. However, such simulation pairs for future scenarios such as SSP5-3.4-OS are not usually
available. The "fLuc" variable provided by some ESMs enables an alternative way to incompletely quantify direct
LUC emissions that include 'deforestation' (biomass loss during deforestation), wood harvest, and the release of
$CO_2$ by harvested wood products, but exclude forest regrowth and legacy soil carbon decay or gains. Three
models, IPSL-CM6A-LR, CNRM-ESM2-1, and UKESM1-0-LL under consideration, provide the variable "fLuc"
(Table 1).
Gridded CMIP6 data, with the exception of the "fLuc" variable, were adjusted by subtracting the long-term pre-
industrial linear trend from the control (piControl) experiment at a grid level. We used the anomalies relative to


the branching year values (indicated in Table S1) for changes in carbon pools and long-term mean piControl
values for changes in carbon fluxes.
**2.3 Methodology**
ESMs do not provide necessary outputs to diagnose the specific carbon fluxes generated from the transitions to
bioenergy crops: 1) they do not treat energy crops explicitly but rather use a generic "crop" vegetation type, itself
being a grass with a higher photosynthesis rate in some models, 2) crops only cover a fraction (tile) of a model
grid box, and 3) the soil carbon pool is usually not split into tiles for each vegetation type in land surface models.
Hence there is no perfect way to diagnose such fluxes. We pragmatically decompose the global $\beta$ and $\gamma$
contributions to the changes in land carbon uptake to the contributions that are LUC- and noLUC-induced by
using three different approaches described below. In all three considered approaches, the $\gamma$ feedback parameter is
estimated from BGC and COU simulations and not from radiatively coupled (RAD) simulations, where only
changes in temperature affect the carbon-cycle processes. First, RAD simulations were not available for the SSP5-
3.4-OS pathway. Second, previous studies suggest that using COU and BGC pair for calculating feedback
parameters may be more representative because using RAD simulations leads to nonlinearity (non-additivity) of
$\beta$ and $\gamma$ feedbacks (Jones et al., 2016b; Schwinger and Tjiputra, 2018; Arora et al., 2020).
In the *"fLuc" approach (1)*, we exploit the "*fLuc*" variable provided by most models in CMIP6. We estimate the
carbon-concentration $\beta$ (GtC ppm$^{-1}$) and carbon-climate $\gamma$ (GtC °C$^{-1}$) feedback parameters using BGC and COU
simulation outputs as described in previous studies (Friedlingstein et al., 2006; Gregory et al., 2009; Jones et al.,
2016; Melnikova et al., 2021):
$\beta = \frac{\Delta C_{BGC}}{\Delta CO_2}$,                                                                                                                              (1)
$\gamma = \frac{\Delta C_{COU} - \Delta C_{BGC}}{\Delta T}$ ,                                                                                                        (2)
where $\Delta C_{BGC}$ and $\Delta C_{COU}$ indicate the changes in the land carbon pool (or cumulative uptake) in BGC and COU
simulations, respectively, and $\Delta CO_2$ and $\Delta T$ (from COU runs) indicate the changes in the $CO_2$ concentration and
global mean surface air temperature (GSAT), respectively, all reported changes being relative to pre-industrial
level (piControl). Because the $CO_2$ atmospheric concentration is not always reported in the model output, we
estimate $\Delta CO_2$ directly from the global $CO_2$ concentration of input4MIP data set which includes the atmospheric
$CO_2$ concentration pathway used in the concentration-driven simulations (Meinshausen et al., 2020). We
performed the calculations using 3-year moving averages.
The global carbon flux, NBP that includes changes in ecosystems both with LUC and noLUC effects, cumulated
over time, approximates the changes in the land carbon pool. Thus, cumulative NBP + fLuc (because NBP and
fLuc have opposite sign conventions with NBP positive sink to land) approximates the changes in the land carbon
pool of noLUC ecosystems. Thus, equations (1) and (2) may be transformed to:
$\beta = \beta_{noLUC} - \beta_{LUC}$,                                                                                                                              (3)
with
$\beta_{noLUC} = \frac{\int NBP_{BGC} dt + \int fLuc_{BGC} dt}{\Delta CO_2}$,                                                                                (4)
and
$\beta_{LUC} = \frac{\int fLuc_{BGC} dt}{\Delta CO_2}$,                                                                                                          (5)
$\gamma = \gamma_{noLUC} - \gamma_{LUC}$                                                                                                                          (6)





with
$\gamma_{noLUC} = \frac{(\int NBP_{COU}dt - \int NBP_{BGC}dt) + (\int fLuc_{COU}dt - \int fLuc_{BGC}dt)}{\Delta T}$,  (7)
and
$\gamma_{LUC} = \frac{\int fLuc_{COU}dt - \int fLuc_{BGC}dt}{\Delta T}$ .  (8)
_In the "cropland threshold" approach (2)_, we divide the global land area into energy-crop-concentrated and
no-energy-crop (not energy-crop-concentrated) grid cells by taking into account their evolution after 2015.
Hurtt et al. (2020) reported that after 2040, cropland areas expanded "mainly due to large-scale deployment
of second-generation bioenergy crops". We carry out a sensitivity study (Text A1) to label the given grid
cell as crop-concentrated if the cropland fraction of the grid cell is larger than a given threshold. In the
sensitivity analysis, we examine a range of post-2015 cropland fraction thresholds of the grid box area and
select the thresholds that best approximate the total cropland area change in 2015–2100 diagnosed by each
ESM. Then, we estimate areal β and γ by using equations (1) and (2) over the energy-crop-concentrated and
no-energy-crop areas. Under this approach, the treatment of LUC and noLUC lands and the attribution of
the LUC effects on the carbon cycle feedback parameters that are relevant to BECCS are both spatially
explicit. The disadvantage of this approach is that by sampling an arbitrary fraction of crop-concentrated
grid-cells, we inevitably omit some carbon changes in cropland or encroach carbon belonging to non-crop
vegetation.
_In the "two simulations" approach (3)_, we performed additional SSP5-3.4-OS scenario simulations by IPSL-
CM6A-LR and MIROC-ES2L. In addition to standard SSP5-3.4-OS and SSP5-3.4-OS-BGC simulations,
we performed simulations in which land use is held constant corresponding to the 1850 usage (SSP5-3.4-
OS-noLUC1850 and SSP5-3.4-OS-noLUC1850-BGC). In addition, using IPSL-CM6A-LR, we performed
simulations with 2040 land cover usage (SSP5-3.4-OS-noLUC2040 and SSP5-3.4-OS-noLUC2040-BGC).
The difference in NBP between simulations with and without LUC indicates LUC emissions, which are
dominated by bioenergy crops area expansion after 2040. The β$_{LUC}$ and γ$_{LUC}$ are estimated as the difference
in β and γ contributions, respectively, between two sets of simulations. Unlike in approaches (1) and (2), the
term LUC here incorporates a carbon source called the "loss of additional sink capacity" (LASC) relative to
the reference years 1850 and 2040 (Gasser and Ciais, 2013; Pongratz et al., 2014). LASC is a change in
carbon flux, or a foregone sink, in response to environmental changes on managed land compared to potential
natural vegetation. The approach (3) accounts for the indirect LUC emissions while the approaches (1) and
(2) do not.
**3 Evaluation and data consistency**
The SSP5-3.4-OS is a concentration-driven scenario based on the implementation of SSP5 in the REMIND-
MAgPIE (Kriegler et al., 2017; Meinshausen et al., 2020). Bauer et al. (2017), Popp et al. (2017), and Riahi et al.
(2017) provide the quantifications, including changes in energy and land use, for the scenario by the IAM. Hurtt
et al. (2020) provided the changes in land use in a coherent gridded format required for ESMs in the Harmonization
of Global Land-Use Change and Management version 2 (LUH2) project. In LUH2, the historical data (up to the
year 2014) based on the History of the Global Environment database (HYDE) and future scenarios (2015–2300)



based on IAM are harmonized to minimize the differences between the end of historical reconstruction and IAM
initial conditions (Hurtt et al., 2020). The harmonization process, however, is expected to result in some
mismatches between LUH2 and the IAM during the early stage of the post-2014 period. First, we check the
consistency of the global and regional cropland and other land-state areas reported by REMIND-MAgPIE, LUH2,
and CMIP6 ESMs. Second, we evaluate global and regional historical LUC estimates by CMIP6 ESMs against
three bookkeeping approaches.

### 3.1 Consistency of cropland area between REMIND-MAgPIE, LUH2, and ESMs

Under the SSP5-3.4-OS pathway, the cropland area increases by 50% from the 2010 level in the 21st century, so
that it reaches $8.1 \times 10^6 \, km^2$ in 2100 (Hurtt et al., 2020). The global cropland area modelled by REMIND-MAgPIE
and downscaled by LUH2 increases due to the expansion of second-generation bioenergy crops. The global
cropland areas by REMIND-MAgPIE and LUH2 are largely consistent with a slightly larger area of crops by
REMIND-MAgPIE till the 2050s (reaching $0.6 \times 10^6 \, km^2$ in the year 2050) and a larger area of crops by LUH2
in 2060 – 2090s (Figure S2). Unlike in the REMIND-MAgPIE, LUH2 simulates a slight reduction of forest area
(by $1.3 \times 10^6 \, km^2$ in 2100 from 2010 level). The differences in the global cropland area between LUH2 and
REMIND-MAgPIE reach $2.9 \times 10^6 \, km^2$ in the year 2060 that is 14% of the total cropland area of $20.7 \times 10^6 \, km^2$
by LUH2 in 2060 (and corresponds to a 43.4% increase from the 2015 level) and may cause additional uncertainty
in estimates of the BECCS area and LUC. Further, ESMs implement the global and regional gridded cropland
fractions following LUH2 and using their own land cover map (Figure S2), with an exception of UKESM1-0-LL
that reports an evolution of the global cropland area smaller than those of other ESMs. This deviation of UKESM1-
0-LL may occur because of its specifications in the treatment of croplands and the model's dry bias (precipitation
deficit) in India and the Sahel (Sellar et al., 2019). While the model used the LUH2 data to prescribe an area
available for crops to grow in, this area will be covered by the crop PFTs only if the model's climate is suitable
for the grass PFTs, otherwise, the area will remain bare soil.
Aside from the deviations in total areas of land cover types between REMIND-MAgPIE, LUH2, and ESMs listed
above, a discrepancy arises from the implementation of LUH2's land cover types to the ESM's plant functional
types (PFTs). Nevertheless, most CMIP6 ESMs produce croplands area consistent with LUH2. However, the other
vegetation classes of LUH2 (e.g., forested lands, non-forested lands, pastures) do not match the PFTs of ESMs
because most ESMs decided to use their own land cover map rather than used the LUH2 one for these ecosystems.
First, spatial distributions of vegetation classes are tightly associated with climate and biogeochemical processes,
and thus, the replacement of the vegetation covers in ESMs would lead to large changes in the model
performances. Second, some models that include dynamic vegetation, like UKESM1-0-LL, predict the vegetation
distribution change, and sometimes the predicted distribution does not coincide with the real one. Besides, the
pastures of REMIND-MAgPIE are translated to two land-use states in LUH2: pastures, and rangelands. While
they are treated predominantly as low-productivity areas in REMIND-MAgPIE, this may not be a case in ESMs,
where pastures and rangelands may correspond to grasslands and perhaps to shrublands (if this land cover exists
in an ESM). We shed light on an issue of inconsistency when translating LUC from IAMs into LUH2 and, then,
into ESMs. Overall, implementation of the LUC scenario of REMIND-MAgPIE to first, LUH2, and then ESMs
leads to a consistency loss of simulated scenario. This problem requires thorough attention in a separate study.



**3.2 Evaluation of land-use change emissions**

The global and regional LUC emissions estimated by ESMs were evaluated against three bookkeeping models for the historical period, namely BLUE (Hansis et al., 2015), HN2017 (Houghton and Nassikas, 2017), and OSCAR (Gasser et al., 2020). The models differ in the spatial units (spatially explicit, country level, region level), parametrization, and process representations (Friedlingstein et al., 2020; Gasser et al., 2020). Unlike other bookkeeping models, OSCAR also reported LASC in LUC estimates but the utilized version did not include peat emissions.

Unlike the difference in NBP between simulations with and without LUC, the "fLuc" variable accounts only for the direct LUC emissions and does not account for all the fluxes reported by bookkeeping models, e.g., forest regrowth and slash and soil organic matter decay, as well as for shifting cultivation and degradation (Houghton and Nassikas, 2017). Thus, its values are expected to be lower. We use an average of multiple realizations when provided by the model teams (details in Table S1). The evaluation targets estimating LUC emissions in "fLuc" and "two simulations" approaches.

We found that ESMs tend to estimate lower global LUC emissions than bookkeeping models by both "fLuc" variable and "two simulations" approaches (Figure 1). This is remarkable in the three tropical regions that dominate global LUC emissions since the 1960s, and particularly South and Southeast Asia (Figure S3). In 1960–2014, on average, bookkeeping models estimate that three tropical regions account for $56.8 \pm 2.3\%$ of global LUC emissions, while ESMs estimate that they account for $35 \pm 10\%$ based on simulations with and without LUC and $40 \pm 15\%$ based on the "fLUC" variable.

LUC emission estimates by MIROC-ES2L (for which only LUC emissions derived from simulations with and without LUC were available) are the most consistent with the estimates of bookkeeping models among considered ESMs (see also Liddicoat et al (2021)). We excluded the estimates of LUC emissions by CNRM-ESM2-1 based on simulations with and without LUC and by UKESM1-0-LL based on "fLuc" from the analysis. CNRM-ESM2-1 estimates much lower LUC emissions derived from simulations with and without LUC than other ESMs, possibly because the CMIP6 version of the model does not include a harvest module, i.e., cropland is modelled as a natural grassland (Séférian et al., 2019) and cropland soils continue to be loaded by harvest inputs. UKESM1-0-LL estimates implausibly low LUC emissions derived from the "fLuc" variable.

The LUC emissions estimated by the two approaches differ remarkably due to inconsistent "fLuc" definitions among models (Gasser and Ciais, 2013). We call for a clearer and more rigorous definition of this variable in future MIPs so that model outputs can be compared on the same basis. As some examples for improvement, we suggest that model teams provide variables contained within "fLuc", e.g., direct deforestation and wood harvest emissions, decomposition flux, as well as indirect emissions, e.g., per each PFT.

**3.3 Evaluation of land-use change emissions from BECCS deployment**

BECCS dominates negative emissions in the SSP5-3.4-OS pathway. We confirmed that BECCS is predominantly deployed in low-carbon uptake areas by comparing the changes in carbon pools and NBP globally and crop-concentrated areas (Figure S4). Because bioenergy crops are deployed in low-carbon uptake areas and they dominate LUC emissions in the 21$^{st}$ century, the NBP over crop-concentrated areas derived by the "cropland threshold" approach approximates global LUC emissions (Figure S5). The comparison of NBP in crop-concentrated grids with the original LUC emissions of the REMIND-MAgPIE IAM scenario confirms a similar



trend between IAM-based global LUC emissions and ESMs-based global temporal NBP changes in the crop-
concentrated areas after 2040. The strong correlation is evident in three ESMs, namely CanESM5, UKESM1-0-
LL, and MIROC-ES2L (correlation coefficient is 0.72 for the 2015–2100 period). The carbon loss in the crop-
concentrated areas over the 21st century period averaged over these three ESMs reaches 37.8 ± 30.3 GtC. Two
models, IPSL-CM6A-LR and CNRM-ESM2-1, however, do not capture the increased carbon loss after 2040
perhaps due to low estimates of LUC emissions from crop expansion (especially, CNRM-ESM2-1) or
overestimated uptake by no-LUC areas (Figures 1, S3). Besides, IPSL-CM6A-LR simulates the lowest ecosystem
carbon pool, especially in soils (Figure S6) that may lead to relatively small LUC-induced carbon losses when
cropland areas expand. Thus, the estimates of LUC impact on carbon cycle feedbacks from IPSL-CM6A-LR and
CNRM-ESM2-1 need to be considered with the above-mentioned caveats.
**4 The impact of LUC on the carbon cycle feedback parameters**
**4.1 Differences in LUC impact on carbon cycle estimated by three approaches**
We use the estimates of the LUC impacts on global $\beta$ and $\gamma$ by IPSL-CM6A-LR and MIROC-ES2L to compare
the three approaches described in sect. 2.3 between each other. The "cropland threshold", unlike the other two
approaches, separates cropland-concentrated and no-crop contributions spatially. Thus, the estimated carbon cycle
feedback parameters are areal cumulative. In the other two approaches, in contrast, the $\beta$ and $\gamma$ are calculated in
each grid cell for both LUC-dominated and noLUC ecosystems, so that carbon change of these two land-use
categories may partly offset each other. For this reason, the carbon cycle feedback parameters, especially $\beta$,
estimated via "cropland threshold" are of smaller magnitudes (sometimes by several times) than those estimated
via the "two simulations since 1850" approach (Figure 2). A larger loss is seen in "two simulations since 1850"
because these simulations include LASC and legacy soil emissions (Figure 2a). Intermediate loss is from "fLUC"
because this approach includes only immediate (direct) carbon loss. Lower carbon losses correspond to "cropland
threshold" that also includes a carbon sink in natural ecosystems over selected grid cells and misses initial carbon
loss, and to "two simulations since 2040" that misses legacy emissions of activities before 2040. The larger carbon
losses in "two simulations since 1850" than in "two simulations since 2040" also reveal the long-term effects of
LUC.
In the case of IPSL-CM6A-LR, the "fLuc" and "two simulations" approaches suggest that a BECCS-related
carbon loss leads to a negative $\beta_{LUC}$, and the "cropland threshold" and "two simulations" approaches suggest that
a carbon loss and climate effects on enhancing soil carbon decomposition leads to a negative $\gamma_{LUC}$. The "cropland
threshold" and "two simulations since 2040" approaches produce similar estimates of LUC impact on cumulative
$\beta$ and $\gamma$ contributions to land carbon because these two methods target the changes in the carbon fluxes particularly
due to cropland expansion for BECCS in the 21st century (Figure 2). MIROC-ES2L that account for gross LUC
emissions (Liddicoat et al., 2021) produce similar estimates of LUC impact by "cropland threshold" and "two
simulations since 1850" approaches (except for differences in $\beta$ explained above).
**4.2 Temporal impacts of LUC on global $\beta$ and $\gamma$**
Figure 3 illustrates the attribution of global $\beta$- and $\gamma$-driven carbon fluxes to LUC- (or crop-concentrated) and no-
LUC (no-crop) ecosystems by five ESMs and three approaches. They show that the large-scale deployment of





bioenergy crops even on low carbon-uptake areas causes a carbon loss from the ecosystem and alters carbon cycle
feedback parameters.
For the "cropland threshold" approach, the majority of ESM simulations, excluding IPSL-CM6A-LR and CNRM-
ESM2-1 (see section 3.3), agree that cropland expansion causes a decrease in global β and that β is negative in
crop-concentrated grids which lose carbon from LUC (Figure 3, Table S2). Cropland expansion for BECCS may
also contribute to the global γ change towards more negative values (larger carbon source). However, these
changes are small in the "cropland threshold" and absent in "fLUC" estimates. We speculate this occurs because
the "fLuc" variable involves only direct LUC changes such as deforestation, wood harvest, and soil carbon decay.
On top of it, earlier findings show that the ESMs misrepresent the amplitude and rate of changes in soil and litter
carbon after LUC (Brovkin et al., 2021).
The γ estimates are in less agreement between ESMs than β, and they are more uncertain when estimated from
BGC and COU simulations, as opposed to RAD runs (Schwinger and Tjiputra, 2018). The LUC carbon losses for
BECCS deployment cannot be overridden by the increased $CO_2$ effects (Figure S7). This causes a decrease in the
global β feedback parameter. Although more studies are needed to confirm the impacts of BECCS-associated
LUC carbon losses on the global γ, the majority of simulations confirm a contribution of LUC ecosystems to the
negative γ. Overall, the three approaches and five ESMs demonstrate that the BECCS expansion under the SSP5-
3.4-OS pathway results in $42.55 \pm 41.08$ GtC loss that corresponds to 12.2% of noLUC β-driven uptake and to an
additional $13.00 \pm 12.27$ GtC loss that corresponds to 14.6% of noLUC γ-driven loss over the 2000–2100 period
(Tables S2, S3).
**4.3 Spatial variation of impacts of LUC on global β and γ**
The spatial variation of β and γ differs among ESMs but the majority agrees on the globally positive β with larger
magnitude in the low latitudes and on the positive and negative γ in the northern high latitudes and low latitudes,
respectively (Figures 4, S8).
The carbon cycle feedback parameters are expected to increase after the peak of $CO_2$ concentration and
temperature. Melnikova et al. (2021) reported that in the SSP5-3.4-OS scenario, the global β and γ parameters
continue to increase even after the peaks of $CO_2$ concentration and temperature at least till the end of the 21$^{st}$
century due to the inertia of the Earth system. After the start of BECCS deployment, the β parameter increases
globally except for the BECCS areas, especially in the tropical region, where it becomes negative (Figure 4).
These differences are apparent in CanESM5, UKESM1-0-LL, and MIROC-ES2L. The positive γ parameter
increases in the high latitudes except for the areas of the eastern part of North America and part of Europe almost
exactly in the areas of BECCS deployment, where the γ is zero or negative (Figure S8). The increase in negative
γ over low-latitudes occurs both in the areas with and without the presence of BECCS. Even though the SSP5-
3.4-OS scenario is designed so that BECCS utilizes low carbon areas to cause the least possible impact on the
carbon sink in unmanaged lands, these BECCS areas lose their β-driven carbon uptake potential but do not escape
γ-driven carbon losses. The spatial variation of β and γ in simulation with and without LUC by MIROC-ES2L
and IPSL-CM6A-LR also demonstrates a decrease in positive β and to a lesser scale an increase in negative γ over
BECCS areas (Figure S9).
We explored the drivers of the β- and γ-driven carbon losses by analysing the changes in the spatial distribution
of ecosystem carbon turnover time $\tau_{eco}$ defined as the ratio of land carbon stock to net primary production (NPP).



Previous studies demonstrated that land-use change is a major driver of $\tau_{eco}$ decrease (Wu et al., 2020; Erb et al.,
2016). We found that the majority of ESMs (with an exception of CNRM-ESM2-1) show an acceleration of carbon
turnover in the areas of BECCS deployment due to LUC-driven ecosystem carbon loss (Figure S10 a-c). The
carbon cycle feedback parameters are directly correlated with $\tau_{eco}$ over the areas of BECCS deployment, which is
apparent in CanESM5, UKESM1-0-LL, and MIROC-ES2L (Figure S10 d, e). LUC causes an ecosystem carbon
loss that drives the acceleration of carbon turnover and results in alteration of β and γ feedback parameters.
The β and γ feedback parameters are assumed to be a pure response to the $CO_2$ concentration and temperature
changes. However, they are also a function of the land cover. In the SSP5-3.4-OS scenario, second-generation
biofuel cropland areas estimated by LUH2 reach nearly 6% of global land (potentially vegetated) area in 2100.
Assigning such vast areas to bioenergy crops - even if they correspond to low-carbon content ecosystems - affects
the land carbon uptake and the global carbon cycle feedback parameters. The decision on the assignment of these
areas for energy crops requires assessment of both the current state of the ecosystem, e.g., the carbon content in
vegetation and soil, and the future potential increase in the carbon uptake via the β and γ feedbacks. The
dependency of global β and γ on LUC should be accounted for in developing future mitigation pathways so that
the benefits of BECCS are not minimized by adverse modulations of carbon cycle feedback parameters.
**5 Conclusion**
In this study, we investigated the impacts of BECCS deployment on the carbon-concentration β and carbon-
climate γ feedback parameters under an overshoot pathway. In a broader sense, the land-cover and land-use
associated differences in the initial conditions of ESMs simulations may influence the estimates of global carbon
cycle feedback parameters even under the idealized pathways. The divergences in the pre-industrial land covers
among ESMs lead to spatial differences in the ecosystem carbon stocks (e.g., ESM with larger forest cover has
larger land carbon pool size). Furthermore, the pre-industrial levels of ecosystem carbon stock vary among models
even for identical land-cover types. The estimated global β and γ feedbacks compromise these land-cover-related
uncertainties. While the β and γ are often compared between ESMs in idealized scenarios (such as 1%CO2
increase), the land-cover impacts have not been discussed. Future studies should address the issue by
benchmarking the sets of idealized experiments with different types of land-cover and land-use changes.
In the evaluation part of this study, we highlighted some inconsistencies in the land-use states and their temporal
transitions between the REMIND-MAgPIE, LUH2, and ESMs. While differences in LUC may arise from
differences in process representations across models and initial conditions, we emphasize that the inconsistencies
should be taken into account in comparative studies of IAMs and ESMs. Further work will be required to address
the issue of the level of inconsistency between the IAMs, LUH2, and ESMs that should be tolerated to have
confidence that ESMs and IAMs describe the same scenario.
We exploit five ESMs and three approaches to show that cropland expansion for BECCS causes a carbon loss
even in low-carbon uptake lands and reduces the future potential increase in the global carbon uptake via LUC
impact on the carbon-concentration and carbon-climate feedbacks. Under the SSP5-3.4-OS, the LUC emissions
from BECCS deployment cause a decrease in global β and contribute to negative γ feedback parameter. The fact
that the impact on β dominates that on γ, which probably reflects the larger role of β-driven carbon uptake than
that of γ-driven loss in the current world and under overshoot pathways (of moderate level).



Our results are consistent with IPCC special report on climate change and land (Shukla et al., 2019) and highlight
the need for considering trade-offs in BECCS deployment and other land-uses but, to some extent, they go beyond
this assessment by considering the implication of carbon cycle feedbacks. Our work shows that areas best suited
for BECCS should also be assessed both in terms of their potential amount of the bioenergy yield and potential
future impact on the $\beta$ and $\gamma$ feedback parameters. Future studies need to further investigate the potential of
BECCS to provide negative carbon emissions with little loss of storage from the $\beta$ and $\gamma$ feedbacks.
**Data availability**
The data from the CMIP6 simulations are available from the CMIP6 archive (https://esgf-
node.llnl.gov/search/cmip6), the LUH2 data from https://luh.umd.edu/data.shtml, and the IIASA database via
https://tntcat.iiasa.ac.at/SspDb/dsd?Action=htmlpage&page=welcome. We obtained LUC emission data of
bookkeeping approaches from the modelling teams and https://dare.iiasa.ac.at/103/ for OSCAR.
**Author Contributions**
O.B., P.Ciais, K.Tanaka, and I.M. initiated the study, all co-authors provided input into developing the study
ideas. I.M. performed data analysis and wrote the initial draft. T.H. (MIROC-ES2L) and P.Cadule (IPSL-CM6A-
LR) performed additional ESM simulations. All authors contributed to writing and commenting on the paper.
**Competing Interests**
The authors have the following competing interests: Roland Séférian is editor of ESD.
**Acknowledgments**
We acknowledge the World Climate Research Programme, which, through its Working Group on Coupled
Modelling, coordinated and promoted CMIP6. We thank the climate modelling groups for producing and making
available their model output, the Earth System Grid Federation (ESGF) for archiving the data and providing
access, and the multiple funding agencies who support CMIP6 and ESGF. We thank Richard Houghton of
Woodwell Climate Research Centerfrom for providing the regional annual fluxes for LUC from HN2017, Eddy
Robertson of Met Office, Vivek Arora of Canadian Centre for Climate Modelling and Analysis for providing
additional information on the LUC implementation in ESMs. The IPSL-CM6 experiments were performed using
the HPC resources of TGCC under the allocation 2020-A0080107732 (project gencmip6) provided by GENCI
(Grand Equipement National de Calcul Intensif). This study benefited from State assistance managed by the
National Research Agency in France under the Programme d'Investissements d'Avenir under the reference ANR-
19-MPGA-0008. Our study was also supported by the European Union's Horizon 2020 research and innovation
programme under grant agreement number 820829 for the "Constraining uncertainty of multi-decadal climate
projections (CONSTRAIN)" project, by a grant from the French Ministry of the Ecological Transition as part of
the Convention on financial support for climate services, by the Ministry of Education, Culture, Sports, Science
and Technology (MEXT) of Japan (Integrated Research Program for Advancing Climate Models, grant no.



JPMXD0717935715) and the Environment Research and Technology Development Fund (JPMEERF20192004)
of the Environmental Restoration and Conservation Agency of Japan. RS acknowledges the European Union's
Horizon 2020 research and innovation programme under grant agreement No 101003536 (ESM2025 – Earth
System Models for the Future). RS acknowledges the support of the team in charge of the CNRM-CM climate
model. Supercomputing time was provided by the Météo-France/DSI supercomputing center.



**Appendix**
**Text A1. Sensitivity study for deriving the crop-concentrated grid thresholds**
Neither IAMs nor ESMs provide BECCS-related LUC emissions. Separating BECCS-related emissions from all
other LUC emissions is virtually impossible due to spatial heterogeneity and many complex factors that affect the
bioenergy crop deployment.
ESMs do not distinguish second-generation bioenergy crops from other crops in CMIP6. Moreover, the cropland
area in ESMs is defined at a sub-grid scale (i.e., on a fraction or tile of a grid box). Because land-use states (e.g.,
forest, crops, pastures) vary in productivity and, thus, carbon uptakes and because modelling teams do not provide
NBP estimates at the sub-grid level, to estimate the area and carbon fluxes of the biofuel crops in ESMs, we
assume that all croplands deployed after the 2040s are for second-generation biofuel crops (Figure A1). We label
the given grid of CMIP6 simulation outputs as crop-concentrated if the cropland fraction of the grid is larger than
a given threshold derived via a sensitivity analysis (Figure A1).

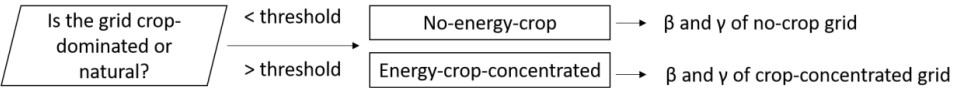


**Figure A1: A schematic presentation of the sensitivity study for estimating the carbon-climate feedback parameters**
**over the energy-crop-concentrated and no-energy-crop grids.**
We examined time-invariant cropland fraction thresholds ranging from 25% to 45% of the grid box area and
selected a range of thresholds that best approximate the change in the total cropland area of each ESM in 2015–
2100 (Figure A2). Here we choose the fitting period of 2015 – 2100 because a shorter period (2040–2100) would
result in a lower threshold during the 2050 – 2060 period with a large global cropland increase. More specifically,
we selected a range of thresholds with a 1%-step so that they intersect at least once either the global cropland area
estimated by ESM itself or LUH2 data set from 2015 to 2100. Although, the selected ensembles of thresholds are
time-invariant, the resultant cropland area increases. We find that for a later period (end of the 21$^{st}$ century), a
higher threshold is required because both the spatial coverage (the number of grid boxes that have crops) and
cropland concentration (a grid fraction of cropland) increases (Figure A2).
We confirmed the spatial distribution of the minimum and maximum selected thresholds of energy-crop-
concentrated grids against sub-grid scale ESM and the LUH2 estimates of cropland area (Figure A3).

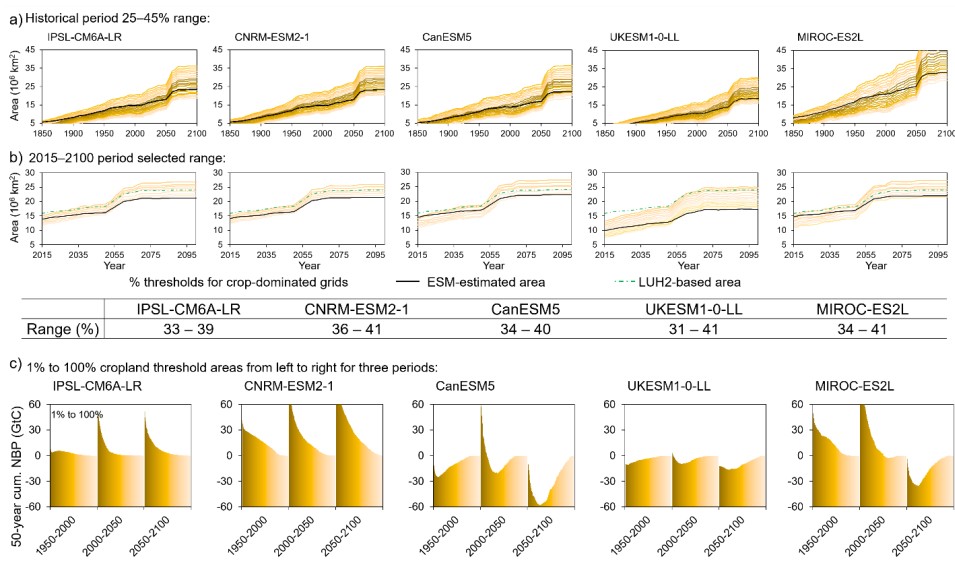

**Figure A2: (a) The cropland-fraction thresholds ranging from 25% to 45% of the grid box area analyzed in the sensitivity study and (b) the selected (resultant) range of thresholds for identifying the energy-crop-concentrated area with the selected range for each ESM indicated in the table. Panel (c) shows the cumulative NBP of the areas corresponding to the range of cropland thresholds from 1 to 100% (left dark to right light color) in three periods.**

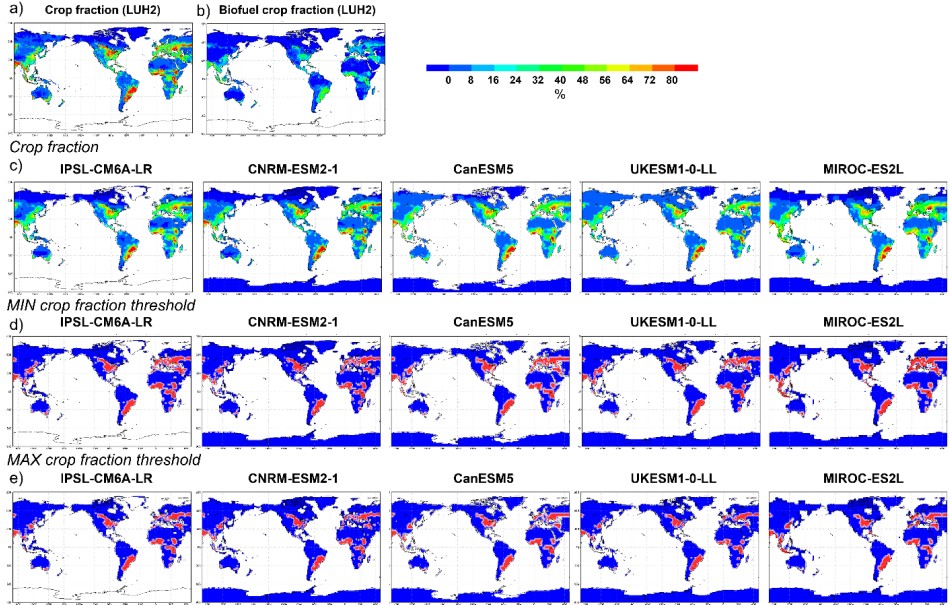

**Figure A3: Spatial variation of (a) grid cropland fraction (b) and second-generation bioenergy cropland fraction by LUH2. Panel (c) shows the spatial variation of grid cropland fraction estimated by CMIP6 ESMs. The spatial variation of the selected (d) minimum and (e) maximum thresholds (that intersect at least once either the global cropland area estimated by ESM itself or LUH2 data set from 2015 to 2100 as shown in Figure A1) for estimating crop-concentrated grids in 2100.**



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



**Tables & figures**

**Table 1. Major characteristics of the Earth system models.**

| ESM* | Reference | Land carbon model | Resolution (land) | Inclusion of "fLuc" | Processes included to "fLuc" |
|---|---|---|---|---|---|
| **IPSL-CM6A-LR** | (Boucher et al., 2020) | ORCHIDEE, br.2.0 | 144 × 143 | Yes | deforestation decomposition |
| **CNRM-ESM2-1** | (Séférian et al., 2019) | ISBA-CTRIP | 256 × 128 | Yes | deforestation decomposition |
| **CanESM5** | (Swart et al., 2019) | CLASS-CTEM | 128 × 64 | No | |
| **UKESM1-0-LL** | (Sellar et al., 2019) | JULES-ES-1.0 | 192 × 144 | Yes (excluded) | deforestation wood harvest decomposition |
| **MIROC-ES2L** | (Hajima et al., 2020) | VISIT-e | 128 × 64 | No | |

*DOIs of simulations by each ESM are provided in Table S1.

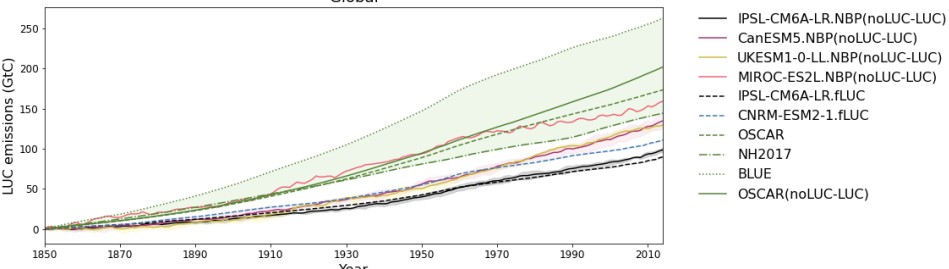

**Figure 1: Evaluation of cumulative global LUC emissions by ESMs against three bookkeeping models. LUC emissions are defined by two methods: 1) the difference in NBP between simulations with and without LUC (solid lines) and 2) the "fLuc" variable provided in CMIP6 (dashed lines). The estimates of bookkeeping approach using OSCAR are shown for cases with (noLUC-LUC) and without (LASC). The range of bookkeeping models is in shaded green.**





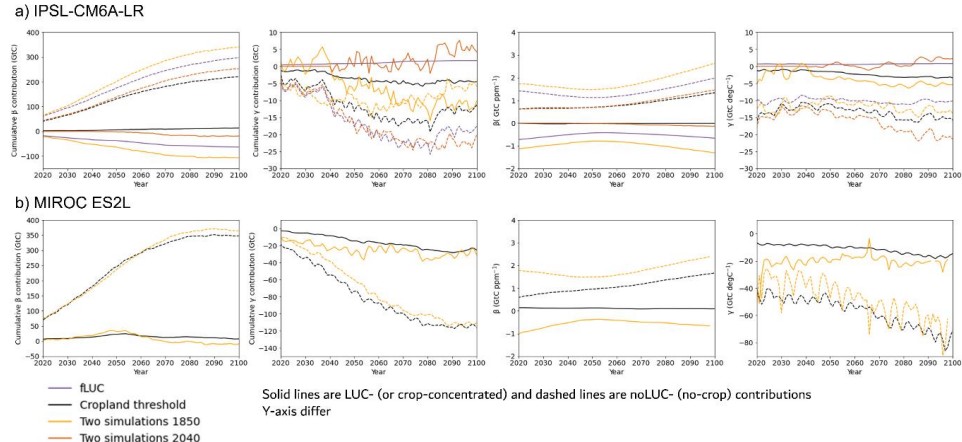

**Figure 2: Differences in the impact of LUC on carbon cycle estimated by three approaches by (a) IPSL-CM6A-LR and (b) MIROC-ES2L. From left to right: cumulative β and γ contributions to land carbon uptake, and global land β and γ feedback parameters in LUC-concentrated (solid lines) and noLUC (dashed lines) ecosystems.**

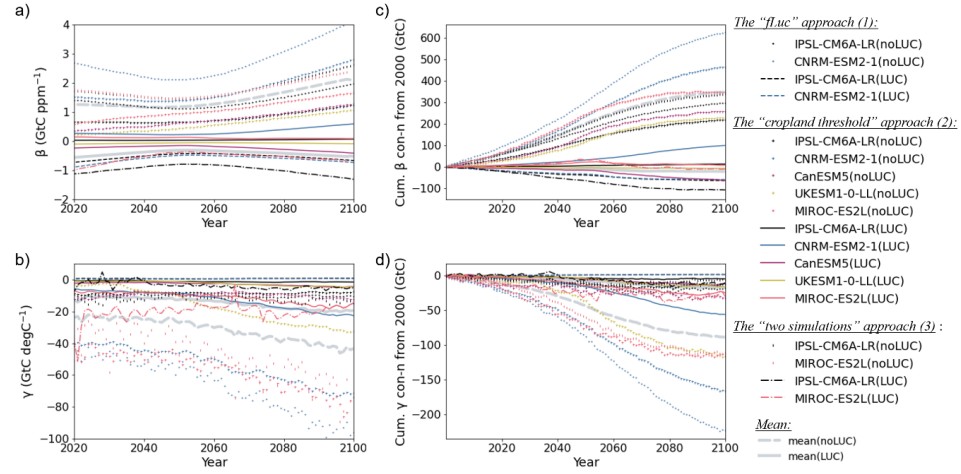

**Figure 3: Global (a) β and (b) γ feedback parameters, and cumulative (c) β and (d) γ feedback contributions to land carbon uptake in LUC or crop-concentrated and noLUC or no-crop ecosystems by three approaches.**



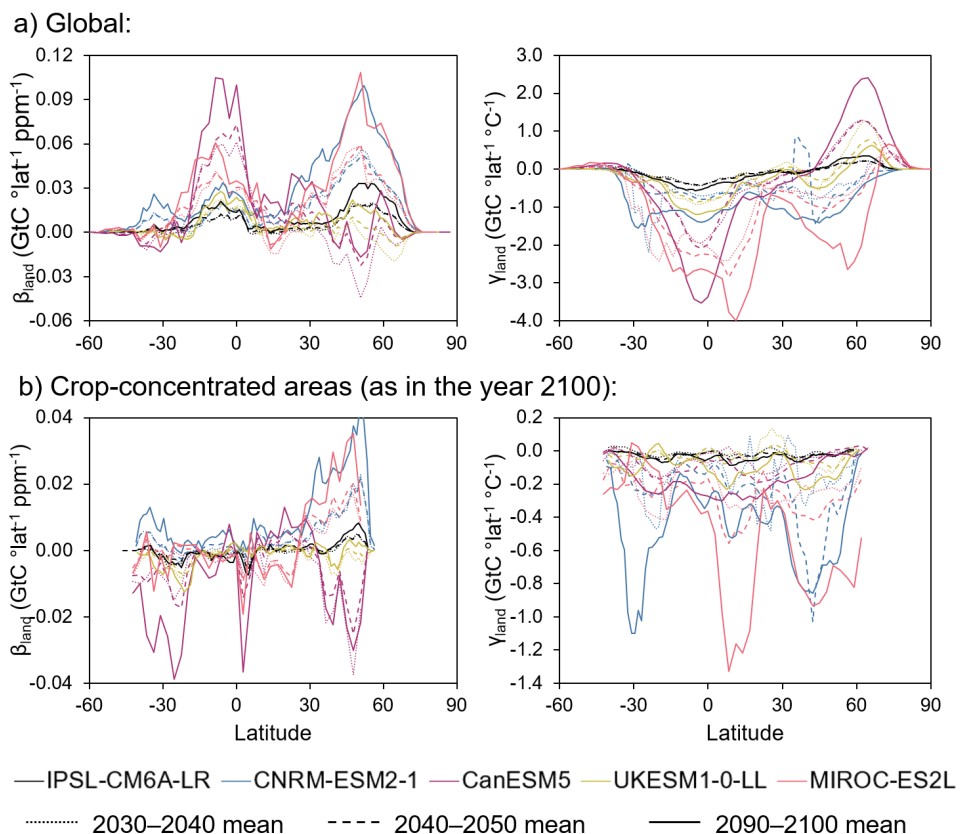

**Figure 4: Latitudinal distributions of the land β and γ feedback parameters (a) globally and (b) in crop-concentrated areas in the SSP5-3.4-OS pathway by five CMIP6 ESMs used in this study. Parameters in crop-concentrated areas are calculated as means of values in the range of cropland thresholds defined in Text A1.**