# Peer review of "Impact of bioenergy crops expansion on climate-carbon cycle"

_Earth System Dynamics, 2021_

## Author Comment (AC1)

We thank the Editor and the Reviewers for their comprehensive comments. Below, we provide a point-by-point reply to each comment. The comments by the Reviewers are in black, and our replies in blue. The changes made in the manuscript are in *italics*.

Comment on esd-2021-72', Vivek Arora, 28 Oct 2021

Major comments

Authors analyze the effect of land use change (LUC) related to bioenergy crops in the SSP5-3.4-OS scenario on the beta and gamma carbon feedback parameters. The manuscript is reasonably well written, especially, as it starts out but then the discussion becomes a little hard to follow. There is good science in the manuscript that needs to be reported but right now it is spread between the main text and the supplementary information that makes it somewhat difficult to follow the story coherently.

Dear Dr. Arora,

Thank you for the comprehensive comments. We believe that following your suggestions greatly improved the manuscript. We revised the storyline of the manuscript. One figure from supplementary information (SI) was moved to the main manuscript and some unnecessary figures were removed from SI.

Other than this, my second major concern is that beta and gamma are just distractions from what is a decent science story. In my opinion, beta and gamma were never designed for scenarios in which CO2 concentrations stabilize or decrease let alone for scenarios with LUC. Recall that beta is given by cumulative change in carbon over land/ocean divided by change in CO2 concentration. Now consider a scenario in which CO2 concentration first rises and then drops back to its pre-industrial value. In such a scenario, cumulative C uptake will be positive but the change in CO2 concentration at the end of the scenario will be zero, yielding a beta value of infinity. As atmospheric CO2 concentration decreases after the peak, beta will keep on increasing since its being divided by a decreasing value of change in CO2 concentration since the pre-industrial times. And, for the same reason gamma is also not meant to be calculated in a scenario with decreasing temperature (which, of course, follows atmos. CO2 concentration). beta and gamma were only meant to compare their magnitude across models and that's also the reason they shouldn't be compared across models if scenarios with LUC and non-CO2 GHGs because the effect of different treatment of LUC and non-CO2 GHGs in models will be lumped into beta and gamma feedback parameters. This is the reason beta and gamma are compared across models from the 1pctCO2 scenario since they are not tainted by other processes that affect atmos-land C exchange.

At the end of the day, the scenario reported in the manuscript is a specified concentration scenario so change in CO2 is same across all models. What's more important is the change in land C (i.e. the numerator term of beta) and that's what the manuscript should focus on? My suggestion is to drop the beta and gamma altogether and just focus on the effect of BECCS through LUC on land C balance.

If you agree with the above reasoning then please also consider not recommending that LUC must be somehow taken into account in the gamma and beta framework. I agree that beta and gamma are functions of land cover but time-invariant land cover. As soon as land cover

changes and LUC emissions are generated then land C balance will change but this has nothing to do with the response of a given land model to CO2 forcing.

We agree with your suggestion to focus on the effect of BECCS through LUC on land carbon balance and not the β and γ feedback parameters. We change the manuscript figures so that they show cumulative carbon flux values in biogochemically-coupled (BGC) and fully coupled (COU) simulations instead of β and γ feedbacks parameters. And the discussion around the figures is changed accordingly. While we removed the discussion related to the β and γ parameters throughout the manuscript, we added a new section at the end to raise the question and discussion on the impact of LUC on β and γ for the reason described below. You brought up two arguments that are 1) "beta and gamma were never designed for scenarios in which $CO_2$ concentrations stabilize or decrease", and 2) "beta and gamma are functions of land cover but time-invariant land cover". We have a slightly different opinion on this.

1) Indeed, your concern on the definition of the β and γ feedback parameters is well founded, as in the case when the $CO_2$ concentration change during ramp-down period goes to zero. But the change in $CO_2$ concentration never goes to zero in the SSP5-3.4-OS before 2300. Because this threshold is not passed, the feedbacks parameters can be safely calculated and thus we consider that we are still applying correctly the mathematical definitions of these feedback parameters. The fact that a metric is ill-defined in a certain domain does not prevent using it in another domain. One example is the airborne fraction that is not defined around zero anthropogenic emissions, but is still widely used. We added a warning to the reader of such limitations.

   *Note, in the case of the overshoot scenarios, when the $CO_2$ concentration and temperature changes during the ramp-down period go to zero, the definitions described in the equation 1 and 2 become invalid. Although because in this study the change in $CO_2$ concentration and temperature never goes to zero in SSP5-3.4-OS before 2300, the feedbacks parameters can safely be calculated, the limitation should be taken into account.*

   Regarding the physical definition of β and γ feedback parameters, applying them in a non-idealized scenario provide another perspective under more realistic evolutions. In our opinion, the β and γ are not just pure theoretical metrics but practical ways to separate the effects of $CO_2$ vs climate change on carbon storage, and they can be used in other scenarios.
   The β and γ feedback parameters are, of course, useful as metrics for inter-model comparison and for the discussion on TCRE in monotonic 1%$CO_2$ increase idealized scenarios. Metrics calculated on idealized scenarios are interesting but calculating them on more socially relevant scenarios, including overshoot ones, may be more relevant for understanding the changes in the carbon cycle. As a specific example, previously Melnikova et al. (2021) applied the β and γ framework to the SSP5-3.4-OS scenario. One of the conclusions of the study was that in the current world and under overshoot pathways of moderate level, the impact on β dominates that on γ. This reflects the larger role of β-driven carbon uptake than that of γ-driven loss for several decades, even during the ramp-down period, when the $CO_2$ concentration decreases and temperature continues to increase.

2) The carbon cycle perturbed by human activities cannot be decoupled from the land cover and LUC because the new land cover would also be influenced by the changed $CO_2$ and

climate locally and globally. We can decouple β and γ neither from the state of the land use, nor from the pre-industrial state of land cover, nor from other model structural parts, leading to a value for equilibrium carbon stock.

There is an interplay between land cover and the model's response to $CO_2$ (and climate) that has been demonstrated mathematically in Gasser & Ciais (2013). The land cover change does impact the response of a given land model to $CO_2$ forcing: that is exactly what the loss of additional sink capacity (LASC) is. Gasser et al. (2020) quantified it to be a foregone sink of about 30 GtC over the historical period. But this value can only increase as future $CO_2$ concentration will be much higher than in the past. Note also that the LASC is discussed in every Global Carbon Budget (but not included for lack of evaluation by complex land models).

On the one hand, discussed in the manuscript, the model intercomparison studies of β and γ feedback parameters under idealized scenario with a time-invariant land cover involve the uncertainty in the β and γ induced by inter-model differences in the land cover.
On the other hand, under the SSP5-3.4-OS scenario, the LUC impacts the hysteresis of the β and γ that has been previously discussed by Melnikova et al. (2021). By utilizing the standard SSP5-3,4-OS simulations and simulations with fixed land cover in 1850 (IPSL-CM6A-LR and MIROC-ES2L) and 2040 (IPSL-CM6A-LR), we show the LUC-driven hysteresis in the β and γ feedback parameters and β- and γ- driven fluxes (Figure R1 that is Figure 6 in the revised manuscript).

[Figure]

Figure R1: The variation of (a) global $β_{land}$ (GtC ppm$^{-1}$) and $γ_{land}$ (GtC °C$^{-1}$), and (b) cumulative over 2000–2300 (for IPSL-CM6A-LR) and over 2000–2100 (for MIROC-ES2L) β- and γ-driven land carbon uptakes with and without LUC. The changes in LUC are given as 9-year moving averages, negative value corresponds to a land sink.

The new section is added as follows.

**5 The carbon cycle feedback framework perspective**

*The $CO_2$ fertilization effect- and climate change-driven changes in the carbon fluxes and storages may be expressed as $\beta$ and $\gamma$ feedback parameters per unit changes in the global atmospheric $CO_2$ concentration ($\Delta CO_2$) and surface air temperature ($\Delta T$), respectively (Jones et al., 2016b; Friedlingstein et al., 2020; Zhang et al., 2021).*

*Here the temperature change is taken as a proxy for the response of the ecosystem carbon storage to climate change. The carbon-concentration $\beta$ (GtC ppm$^{-1}$) and carbon-climate $\gamma$ (GtC °C$^{-1}$) feedback parameters can be estimated using BGC and COU simulation outputs (Friedlingstein et al., 2006; Gregory et al., 2009; Jones et al., 2016; Melnikova et al., 2021; Zhang et al., 2021):*

$$\beta = \frac{\Delta C_{BGC}}{\Delta CO_2}, \tag{1}$$

$$\gamma = \frac{\Delta C_{COU} - \Delta C_{BGC}}{\Delta T}, \tag{2}$$

*where $\Delta C_{BGC}$ and $\Delta C_{COU}$ indicate the changes in the land carbon pool (or cumulative uptake) in BGC and COU simulations, respectively, and $\Delta CO_2$ and $\Delta T$ (from COU runs) indicate the changes in the global $CO_2$ concentration and mean surface air temperature, respectively, all reported changes being relative to pre-industrial level (piControl).*

*The $\beta$ and $\gamma$ feedback parameters / metrics are often compared between ESMs in idealized scenarios (such as 1%$CO_2$ increase), and the parameter values are assumed to be a pure response to the $CO_2$ concentration and temperature changes. Applying this framework to non-idealized and more socially relevant scenarios provides another perspective for understanding the changes in the carbon fluxes under more realistic evolutions. Previously, Melnikova et al. (2021) applied the $\beta$ and $\gamma$ framework to the SSP5-3.4-OS scenario and showed an amplification of the feedback parameters after the $CO_2$ concentration and temperature peaks due to inertia of the Earth system. Here we performed an estimation of the $\beta$ and $\gamma$ feedback parameters to investigate the impacts of the LUC on the behavior of the feedback parameters.*

*Note, in the case of the overshoot scenarios, if the $CO_2$ concentration and temperature changes during the ramp-down period went to zero, the definitions described in the equation 1 and 2 would become invalid. Although because in this study the change in $CO_2$ concentration and temperature never goes to zero in SSP5-3.4-OS before 2300, the feedbacks parameters can safely be calculated, the limitation should be taken into account.*

*The land carbon uptake and the $\beta$ and $\gamma$ feedback parameters are affected by LUC, they are lower in the simulations with LUC (Figure 6). Moreover, the difference in the $\beta$ parameter estimated by IPSL-CM6A-LR in simulations with LUC and without LUC after year 2040 suggests that even only LUC for bioenergy crops expansion affects the hysteresis behaviour of the carbon cycle feedback parameters under declining $CO_2$ concentration and temperature.*

*To date, the LUC impacts on carbon cycle have not been included into the $\beta$ and $\gamma$ feedback framework, and the LUC emissions are discussed as an anthropogenic forcing separately from the feedbacks of land ecosystems to the changed $CO_2$ and climate. However, the $\beta$ and $\gamma$ parameters can be decoupled neither from the state of the land use, nor from the pre-industrial state of land cover, nor from other model structural parts, leading to a value for equilibrium carbon stock. There is an interplay between land cover and the model's response to $CO_2$ (and climate) that has been demonstrated mathematically in Gasser & Ciais (2013) and defined as LASC. Gasser et al. (2020) quantified it to be a foregone sink of about 30 GtC over the historical period. But this value can only increase as future $CO_2$ will be much higher than in the past.*

*In a broader sense, the land-cover and land-use associated differences in the initial conditions of ESMs simulations influence the estimates of global carbon cycle feedback parameters even under idealized pathways. The divergences in the pre-industrial land covers among ESMs lead to spatial differences in the ecosystem carbon stocks (e.g., ESM with larger forest cover has larger land carbon pool size). Furthermore, the pre-industrial levels of ecosystem carbon stock vary among models even for identical land-cover types. The estimated global β and γ feedback values involve these land-cover-related uncertainties. Future studies should address the issue by benchmarking the sets of idealized experiments with different types of land-cover and land-use changes.*

This manuscript has good science to convey the overall message that the BECCS scenarios come at a cost of increased crop area with the associated LUC emissions. In this context, it also becomes important to check that the LUC emissions in the original IAM are not too different from that in the ESMs. Of course, the caveat is that (as you already mention) land use change information gets somewhat distorted as it goes from IAM -> LUH -> ESMs.

The other subtle point, which I am not sure about, is that these enhanced LUC emissions to account for BECCS must be the part of total carbon budget calculations in the IAM scenario. Unless I missed this, I can't recall seeing a comparison of LUC emissions from ESMs with those from the IAM. So the IAM group who designed the scenario must have taken into account that the benefit of BECCS more than compensates for increased LUC emissions. That's why CO2 goes down after its peak in this scenario.

We agree with this point. The IAMs indeed design the scenario in a way that the benefits of BECCS exceed the carbon losses from LUC. However, the ability of IAM to accurately estimate LUC emissions including legacy emissions and long-term consequences is questionable. REMIND-MAgPIE estimates lower LUC emission compared to ESMs (Figure R2). We added this figure to SI (now Figure S2) and the following text to Section 3.3.
*The increased LUC emissions to account for BECCS are a part of total carbon budget calculations in the IAM scenario. We compared LUC emissions by different approaches using ESMs with LUC of REMIND-MAgPIE (Figure S2). While the IAMs design the scenario in a way that the benefits of BECCS exceed the carbon losses from LUC, the ability of IAM to accurately estimate LUC emissions including LASC is questionable. In SSP5-3.4-OS scenario, REMIND-MAgPIE estimates lower LUC emission compared ESMs.*

[Figure]

Figure R2: Comparison of (a) annual and (b) cumulative from year 2040 global LUC emissions by ESMs (by three approaches) against REMIND-MAgPIE under SSP5-3.4-OS scenario. "LUCcrop" indicates LUC emissions estimated via the "cropland threshold" approach. The changes in LUC are given as 9-year moving averages, negative value corresponds to a land sink.

Minor comments

Abstract. The last sentence of the abstract is too long. Please consider rewording it into two smaller sentences. Also its unclear what "so as to limit the reductions of the CO2 fertilization effect" means in this sentence.

We changed the abstract according to your major comment and reworded the last sentence and divided it into two shorter ones.

*The dependency of the land carbon uptake on LUC is strong in the SSP5-3.4-OS scenario but it also affects other SSP scenarios and should be taken into account by the IAM teams. Future studies should further investigate the trade-offs between the carbon gains from the bioenergy yield and losses from the reduced $CO_2$ fertilization effect-driven carbon uptake where BECCS is applied.*

Line 40. "In BECCS, atmospheric CO2 is captured via photosynthesis and fixed into plant biomass". BECCS or not, photosynthesis always captures C and fixes it into plant biomass. Please consider rewording.

This is, indeed, not necessary. Revised to:

*In BECCS, atmospheric CO2 is captured from biomass growth, and the harvested biomass is then converted into bioenergy or ...*

Line 77. "so that the effects of LUC on these parameters are overlooked". This is by design in my opinion.

We removed this phrase as we revised the discussion on the carbon cycle feedback parameters.

Line 89. "include the expansion of second-generation bioenergy crops (for BECCS) at the cost mainly of pasture lands". I have always struggled with pastures. Pasture is not a land cover but rather a land use. Assuming all pastures are grasslands is an incorrect assumption. Are you able to shed any light on how ESMs treat pastures? I know, CanESM5 doesn't treat pastures at all due to this ambiguity in its definition.

Thank you for this comment. A motivation to investigate the loss of information while translating IAMs to LUH2, and then, to ESMs was partly based on the fact that some land use (and not land cover) tiles of LUH2 do not correspond accurately to land covers of ESMs, in particular, pastures and rangelands. In the manuscript, we referred to the description of the land-use change scenario design by Hurtt et al. (2020). But the analysis of the LUH2 data showed that not only pastures but also rangelands were used for the expansion of the croplands. The definitions of the pastures and rangelands may be sometimes ambiguous. In LUH2 they are both grazing land that differ in the aridity and population density indices. Even more ambiguous is the way pasture vs rangelands are interpreted by ESMs. A pasture created upon a forest leads to complete deforestation and large carbon losses. A rangeland created upon a natural ecosystem can leave most trees intact in the real-world (a small carbon loss) whereas if an ESM would treat this land use change like the conversion to a pasture the simulated carbon loss will be coarsely overestimated.

Then, we actually requested the modelling groups to provide more information on how the pastures and rangelands of LUH2 are treated. The summary of answers is in the table below.

| ESM | LUH2 pastures and rangelands |
| --- | --- |
| IPSL-CM6A-LR | Pastures correspond to grass PFTs, rangelands – natural PFTs |

| | |
|---|---|
| CNRM-ESM2-1 | Pastures correspond to grasslands, rangelands – to shrubs |
| CanESM5 | Not treated. Can be grasslands or shrubs |
| UKESM1-0-LL | Pastures are managed grasslands; rangelands correspond to natural PFTs |
| MIROC-ES2L | The "closed pasture" and "rangeland" – natural vegetation, can be grasses or shrubs, that get impact from grazing pressure |

There is a large variation of how the ESMs treat pastures and rangelands. We added this information to Table 1 and the following discussion to the manuscript:

*Some ESMs do not distinguish pastures and rangelands because of the ambiguity in their definitions. Likewise, the SSP5-3.4-OS scenario involves a large-scale second-generation bioenergy crops whose benefit is the capability to grow in so-called "marginal" lands (Krause et al., 2018). The ambiguity and inconsistency in the definition of land-use and land-cover tiles between IAM, LUH2 and ESMs may have implications to the interpretation of the scenario. We shed light on an issue of inconsistency when translating LUC from IAMs into LUH2 and, then, into ESMs. Overall, implementation of the LUC scenario of REMIND-MAgPIE to first, LUH2, and then ESMs leads to a consistency loss of simulated scenario during the harmonization process. The land cover representation in ESMs is model-dependent and different from the IAM and LUH2 mainly because of ambiguity of land-use and land-cover tiles definitions. This problem requires thorough attention especially in ESMs and IAMs intercomparison studies.*

Equation (5). Note that in this eqn. beta_LUC depends on f_LUC and change in CO2. Since f_LUC has nothing to do with CO2 (it depends on externally prescribed change in land cover) in my mind bringing in LUC into the beta and gamma framework doesn't make sense.

We removed this part of the discussion as we do not decompose the β and γ parameters to LUC and noLUC in the revised version of the manuscript.

Line 156. "In the sensitivity analysis, we examine a range of post-2015 cropland fraction thresholds of the grid box area and select the thresholds that best approximate the total cropland area change in 2015–2100 diagnosed by each ESM". Were these threshold ESM specific?

These thresholds are ESM-specific. The ranges for each ESM were derived in the sensitivity analysis and are provided in the table A2. We also added a clarification to the main text "the (ESM-specific) thresholds".

Line 180. " … provide the quantifications, including changes in energy and land use, for the scenario by the IAM". The phrase "for the scenario by the IAM" is unclear.

We changed the text to:
*Bauer et al. (2017), Popp et al. (2017), and Riahi et al. (2017) provide additional details on the changes in energy and land use.*

Section 3.1. I think Figure S2 belongs in the main text.

We moved Figure S2 (now Figure 1) to the main text.

Also, in this section lines 191-200 are confusing and it seems at some places change is reported as absolute value. For example, on line 191 it's mentioned "the cropland area increases by 50% from the 2010 level in the 21st century, so that it reaches $8.1 \times 10^6$ km2 in

2100". The present day cropland area in LUH is around 15 million km2. So how can it increase by 50% from 2010 and still be 8.1 million km2.

*We corrected the sentence, it was indeed erroneous before. Now it reads:*
*Under the SSP5-3.4-OS pathway, the cropland area increases by $8.1 \times 10^6$ km$^2$ (~50%) from the 2010 level in the 21st century to 2100 (Hurtt et al., 2020).*

The issue around transferring land use change information along this chain IAM -> LUH -> ESMs has been raised in the manuscript but it appears to suggest ESMs do not do a proper job. In fact, the problem is that land cover representation in models is very subjective and different from what IAMs and LUH does. Inevitably some information is lost in translation. Perhaps this can be made more clear.

We made changes, also explained in our response to your major comment above, as follows:
*Overall, implementation of the LUC scenario of REMIND-MAgPIE to first, LUH2, and then ESMs leads to a consistency loss of simulated scenario during the harmonization process. Further, the land cover representation in ESMs is subjective and different from the IAM and LUH2 mainly because of ambiguity in the correspondence between land-use and vegetation type definitions. This problem requires thorough attention especially in ESMs and IAMs intercomparison studies.*
*...*
*In the evaluation part of this study, we highlighted some inconsistencies in the land-use states and their temporal transitions between the REMIND-MAgPIE, LUH2, and ESMs. These differences arise from differences in process representations and initial conditions, as well as land-use and land-cover tiles definitions across models.*

Line 215. "… the predicted distribution does not coincide with the real one". What do you mean by "the real one".

Changed to "with the one prescribed by LUH2"

Line 252. "… we suggest that model teams provide variables contained within "fLuc"". Consider replacing this by " … we suggest that model teams provide explicit detail of processes that contribute to "fLuc".

Replaced.

Section 4. Lines 273 to 294 are very difficult to follow. The fact that there are 8 lines in each panel of Figure 2 doesn't help either. The figures need to be simplified in somewhat. Perhaps just show the range (as shaded region) and the mean.

We revised the text and simplified the figure (now Figure 3), reducing the number of panels. In the improved figure we show only the cumulative carbon fluxes in BGC and COU simulations and not the carbon cycle feedbacks parameters. The text is now also simplified as there is no comparison on the feedback parameters.

Line 306. "On top of it, earlier findings show that the ESMs misrepresent the amplitude and rate of changes in soil and litter carbon after LUC". Please consider replacing the word "misrepresent" with "do not realistically". Also the phrase "the amplitude and rate of changes in soil and litter carbon" is unclear. Please consider rewording this.

We changed accordingly and updated the reference:
*...the ESMs do not realistically represent the dynamics of soil and litter carbon after LUC (Boysen et al., 2021).*

Line 359. "The estimated global β and γ feedbacks compromise these land-cover-related uncertainties". Please reword this sentence. I am not sure what "compromise" actually means in this sentence.

We removed the sentence as it is no more relevant in the revised version of the manuscript.

Figure 3 is also very hard to follow with 20 something lines in each panel. You have to find a way to simplify this information.

We simplified the figure by showing the mean and SD of all data (now Figure 4). We moved Figure 3 to SI (now Figure S4) for the readers who require more details.

References

Gasser, T. and Ciais, P.: A theoretical framework for the net land-to-atmosphere $CO_2$ flux and its implications in the definition of "emissions from land-use change", Earth Syst. Dynam., 4, 171–186, https://doi.org/10.5194/esd-4-171-2013, 2013.

Gasser, T., Crepin, L., Quilcaille, Y., Houghton, R. A., Ciais, P., and Obersteiner, M.: Historical $CO_2$ emissions from land use and land cover change and their uncertainty, Biogeosciences, 17, 4075–4101, https://doi.org/10.5194/bg-17-4075-2020, 2020.

Hurtt, G. C., Chini, L., Sahajpal, R., Frolking, S., Bodirsky, B. L., Calvin, K., Doelman, J. C., Fisk, J., Fujimori, S., Goldewijk, K. K., Hasegawa, T., Havlik, P., Heinimann, A., Humpenöder, F., Jungclaus, J., Kaplan, J., Kennedy, J., Kristzin, T., Lawrence, D., Lawrence, P., Ma, L., Mertz, O., Pongratz, J., Popp, A., Poulter, B., Riahi, K., Shevliakova, E., Stehfest, E., Thornton, P., Tubiello, F. N., van Vuuren, D. P., and Zhang, X.: Harmonization of Global Land-Use Change and Management for the Period 850-2100 (LUH2) for CMIP6, Geosci Model Dev., 1–65, https://doi.org/10.5194/gmd-2019-360, 2020.

Melnikova, I., Boucher, O., Cadule, P., Ciais, P., Gasser, T., Quilcaille, Y., Shiogama, H., Tachiiri, K., Yokohata, T., and Tanaka, K.: Carbon cycle response to temperature overshoot beyond 2 °C – an analysis of CMIP6 models, Earth's Future, 9, e2020EF001967, https://doi.org/10.1029/2020EF001967, 2021.

---

## Author Comment (AC2)

We thank the Editor and the Reviewers for their comprehensive comments. Below, we provide a point-by-point reply to each comment. The comments by the Reviewers are in black, and our replies in blue. The changes made in the manuscript are in *italics*.

'Comment on esd-2021-72', Anonymous Referee #2, 11 Dec 2021

Major comments:

The manuscript focuses on an important scientific problem and draws some enlightening conclusions. The authors estimated the impacts of large-scale land-use change (LUC) on the carbon cycle feedbacks under the Shared Socioeconomic Pathway (SSP) overshoot scenario. They used five ESMs of CMIP6 to estimate the global $\beta$ and $\gamma$ contributions to the changes in land carbon pools in LUC/noLUC areas and found that BECCS areas lose their $\beta$-driven carbon uptake potential but do not escape $\gamma$-driven carbon losses even though the SSP5-3.4-OS scenario is designed for bioenergy crops expansion to utilize already low-carbon areas.

Thank you for the positive review and comprehensive comments that helped to improve the manuscript.

However, the following issues need to be figured out before the manuscript is published:

1. It is difficult for me to understand the biophysical meaning of a negative $\beta$ value. From the perspective of the land and ocean reservoirs, $\beta$ is positive, and $\beta$-feedback reduces the impact of $CO_2$ emissions on atmospheric $CO_2$ concentrations and then global warming (*Zhang X, Wang Y P, Rayner P J, et al. A small climate-amplifying effect of climate-carbon cycle feedback[J]. Nature communications, 2021, 12(1): 1-11*). When the decline of carbon uptake ($\Delta C_{BGC}$) is mainly driven by LUC rather than the change in atmospheric $CO_2$ concentration, is it still appropriate to use $\Delta C_{BGC}$ to calculate the feedback of land carbon uptake to the change in $CO_2$ concentration? Please clarify this.

   Indeed, the negative $\beta$ value would indicate that the changes in carbon pool are dominated by LUC rather than by the $CO_2$ fertilization effect. In the revised manuscript, discuss the LUC impact on the carbon uptake in terms of the cumulative carbon fluxes rather than the $\beta$ and $\gamma$ feedback parameters, e.g., as below.

   *The losses from LUC surpass the benefits from the $CO_2$ fertilization effect, so that the LUC ecosystems become a carbon source to the atmosphere during the study period.*

   As it is impossible to decouple the carbon cycle response to $CO_2$ and climate from the LUC state due to the differences in the potential impacts of $CO_2$ and climate on the old and new land cover, we introduce a new Section 5 to the manuscript where we make a discussion using the $\beta$ and $\gamma$ feedback framework. Here we use terminology more carefully, so that we instead of decomposing the feedback parameters to the LUC and noLUC contributions, we explicitly discuss $\beta$ and $\gamma$ of the simulations with and without LUC.

   *The land carbon uptake and the $\beta$ and $\gamma$ feedback parameters are affected by LUC, they are lower in the simulations with LUC (Figure 6). Moreover, the difference in the $\beta$ parameter estimated by IPSL-CM6A-LR in simulations with LUC and without LUC after year 2040 suggests that LUC for bioenergy crops expansion affects the hysteresis behaviour of the carbon cycle feedback parameters under declining $CO_2$ concentration and temperature.*

2. The result section is not easy to read. Although the results and discussion can appear in the same section, they should be as separated as possible. It is suggested that the result comparison between different methods, data, and studies should be placed at the end of the section.

   We revised the manuscript to improve its flow. In section 4.3, particularly, we placed the comparison between studies at the end.

3. The line charts (Figure 2, 3, and 4) need to be simplified. There are so many lines in each subfigure that readers can not clearly distinguish all lines and colors.

4. We simplified all figures. Note that we still kept the lines in Figure 2 (now 3) because its purpose was to evaluate the three approaches, although we reduced the number of panels in that figure from eight to two. In the improved figure we show only the cumulative carbon fluxes in BGC and COU simulations and not the carbon cycle feedbacks parameters.

Minor comments:

1. Line 67~70: Please divide this sentence into two sentences.

   Divided accordingly (now in the Section 5.1).

2. Line 102: Please give the full name of "fLuc", such as forest land-use change.
   We added its full CMIP6 definition that is "net carbon mass flux into atmosphere due to land-use change". Here "f" refers to a flux.

3. Line 191~192: Please modify this sentence. For example: Under the SSP5-3.4-OS pathway, the cropland area increases by $8.1\times106$ km$^2$ (~50%) from the 2010 level in the 21st century to 2100 (Hurtt et al., 2020).
   Thank you, corrected accordingly.

4. Line 198~200: It is suggested to revise this sentence like "*global cropland area in A dataset is larger/less than in B dataset by X km$^2$ in XXXX, and ~* ".
   Changed to

   "*The global cropland area in LUH2 is less than in REMIND-MAgPIE by $0.3 \times 10^6$ km$^2$ in 2015, and larger by $2.9 \times 10^6$ km$^2$ in 2060…*"

5. Line 276: Please add [*under the "cropland threshold" approach*] at the end of the sentence.
   Added.

6. Line 675: It is recommended to keep only the average and range of land carbon uptake in LUC and noLUC in Figure 3.
   We simplified the figure by showing the mean and SD of all data (now Figure 4).